# Red Blood Cell Omega-6 Fatty Acids and Biomarkers of Inflammation in the Framingham Offspring Study

**DOI:** 10.3390/nu17132076

**Published:** 2025-06-22

**Authors:** Heidi T. M. Lai, Nathan A. Ryder, Nathan L. Tintle, Kristina H. Jackson, Penny M. Kris-Etherton, William S. Harris

**Affiliations:** 1Fatty Acid Research Institute, Sioux Falls, SD 57106, USA; nar@faresinst.com (N.A.R.); nlt@faresinst.com (N.L.T.); kristina@omegaquant.com (K.H.J.); pmk3@psu.edu (P.M.K.-E.); wsh@faresinst.com (W.S.H.); 2OmegaQuant Analytics, Sioux Falls, SD 57106, USA; 3Department of Internal Medicine, Sanford School of Medicine, University of South Dakota, Sioux Falls, SD 57105, USA; 4Department of Nutrition, The Pennsylvania State University, State College, PA 16802, USA

**Keywords:** linoleic acid, arachidonic acid, inflammatory markers, *C*-reactive protein, interleukin-6, intercellular adhesion molecule-1, monocyte chemoattractant protein-1, P-selectin, osteoprotegerin

## Abstract

Background/Objectives: Chronic inflammation is recognized as an important risk factor for a variety of health disorders. Omega-6 polyunsaturated fatty acids (*n*-6 PUFAs), particularly linoleic (LA) and arachidonic acid (AA), have been shown to be either pro- or anti-inflammatory, and researchers have advocated both for and against reducing their dietary intake. This study sought to correlate the levels of ten inflammation-related biomarkers across multiple pathways with red blood cell (RBC) membrane levels of the major dietary and circulating *n*-6 PUFAs. Methods: We included 2777 participants (mean age: 66 ± 9 years, 54% women, 9.8% minorities) from the Framingham Offspring and minority-enriched Omni cohorts, and calculated partial correlation coefficients. Results: After multivariable adjustment, RBC LA was inversely correlated (all *p* ≤ 0.05) with five markers of inflammation, receptors, or pathways: *C*-reactive protein (*r* = −0.06); soluble interleukin-6 (*r* = −0.15); intercellular adhesion molecule-1 (*r* = −0.09); monocyte chemoattractant protein-1 (*r* = −0.07); and P-selectin (*r* = −0.07). RBC AA was inversely correlated (all *p* ≤ 0.05) with soluble interleukin-6 (*r* = −0.10); intercellular adhesion molecule-1 (*r* = −0.14); monocyte chemoattractant protein-1, and (*r* = −0.06); and osteoprotegerin (*r* = −0.07). Lipoprotein-associated phospholipase-A2 mass and activity, urinary isoprostanes, and tumor necrosis factor receptor-2 were not significantly correlated with LA or AA. Conclusions: In our large community-based study, we observed weak but statistically significant inverse associations between several types of inflammatory biomarkers with RBC *n*-6 PUFAs. Our findings do not support the hypothesis that omega-6 fatty acids are pro-inflammatory.

## 1. Introduction

The roles that dietary fats play in the etiology of multiple chronic diseases, particularly cardiovascular disease (CVD), are topics of ongoing controversy. The only class of dietary fatty acids (FAs) that is universally believed to be unhealthy is the industrially produced trans FAs [1]. The convention of grouping FAs into classes by carbon chain length and saturation does not imply similar biochemical effects and physiological functions. For example, some dietary saturated FAs (e.g., palmitic acid) have adverse effects on serum cholesterol levels while others in the same family are neutral (e.g., stearic acid) [2,3,4]. The circulating levels of oleic acid, a major component of olive oil (and thus the Mediterranean diet) and also produced through de novo lipogenesis [5], have been positively associated with the risk for depression [6], CVD [5,7], and total mortality [5,7], bringing into question their cardioprotective reputation [5]. Because some clinical trials have not demonstrated a reduction in the CVD risk with marine-derived omega-3 polyunsaturated fatty acids (*n*-3 PUFAs), eicosapentaenoic acid (EPA) and docosahexaenoic acid (DHA) once believed to be cardioprotective [8] are now viewed by some [9] but not all [10,11] as being ineffective.

Relative to health benefits, the most controversial dietary FAs are the omega-6 (*n*-6) PUFAs. Some scientists point to the potential harms of arachidonic acid (AA, C20:4n6) as it is the precursor to multiple pro-inflammatory mediators [12]. AA can be synthesized from the major dietary and tissue *n*-6 PUFA, linoleic acid (LA, C18:2n6). Major food sources of LA in the Western diet are seed oils (e.g., soybean, corn, cottonseed oils) and, to a certain extent, chicken (via their feed) [13], and these have, in recent years, become the focus of much adverse publicity owing to both their “processed” nature and LA content. Some researchers have called for a marked reduction in LA intake, based strongly on the view that LA is pro-inflammatory [12,14,15]. The concern about LA and seed oils is illustrated by the recent Cleveland Clinic Health Newsletter story entitled, “Seed Oils: Are they actually toxic [16]?”.

In contrast, the American Heart Association (AHA) in 2009 issued a Science Advisory that reviewed the role of *n*-6 PUFAs in CVD and recommended that *n*-6 PUFA intake should be at least 5% to 10% of energy. The Advisory concluded, “To reduce *n*-6 PUFA intakes from their current levels would be more likely to increase than to decrease risk for coronary heart disease (CHD)” [17]. In 2012, Johnson and Fritsche conducted a systematic review of the effects of LA on inflammatory biomarkers and stated: “We conclude that virtually no evidence is available from randomized, controlled intervention studies among healthy, non-infant human beings to show that addition of LA to the diet increases the concentrations of inflammatory biomarkers [18]”. Two major studies that pooled the de novo analyses of individual data from multiple cohorts found circulating LA levels to be inversely associated with the risk for CVD [19] and for type 2 diabetes mellitus (T2DM) [20], and a recent report from the UK Biobank found strong inverse associations between plasma LA levels and total and cause-specific mortality [21].

With regard to the other major *n*-6 PUFA, when AA was fed to mice with experimental colitis, there was no increase in bowel inflammation compared with the oleic acid-rich control diet, and the AA-treated mice had less diarrhea [22]. Supplementation trials with AA in humans have not observed any increase in pro-inflammatory metabolites [23,24]. Finally, although higher AA levels in adipose tissue were directly associated with the risk for CHD events in the Diet, Cancer and Health study [19], when pooled together with 29 other cohorts with data on in vivo AA levels, this FA was unrelated to the risk for CVD [19]. Similarly, in a paper from the Cardiovascular Health Study, plasma phospholipid AA was directly association with the risk for incident T2DM, but when combined with similar data from 19 other cohorts, the overall relationship was null [20].

The purpose of the present study is to further explore the hypothesis that LA and/or AA are pro-inflammatory by examining the cross-sectional association between the red blood cell (RBC) membrane levels of LA and AA and ten biomarkers representing different phases and pathways of inflammation in a large, community-based sample. Based on our reading of the prior literature, we hypothesized that, overall, LA would be inversely correlated with several inflammatory biomarkers, and that no relationship would be found with AA. This study took the same approach as Fontes et al. [25] who reported the associations between the RBC *n*-3 PUFA levels and these ten inflammatory biomarkers in this cohort.

## 2. Materials and Methods

### 2.1. Study Sample

The Framingham Heart Study is a longitudinal community-based cohort study that was initiated in 1948. The selection criteria for the Framingham Offspring cohort and the Framingham Omni cohort have been described previously [26,27] (https://www.nhlbi.nih.gov/science/framingham-heart-study-fhs, accessed on 28 April 2025). Briefly, adult children of the original cohort were recruited in 1971 into the Framingham Offspring cohort. To reflect the increased diversity of the community as the population has changed in Framingham, the ethnic/racial minority Omni cohort was recruited in 1994 [28]. We evaluated Framingham Offspring participants (*n* = 2985) who attended their eighth examination cycle (2005–2008) and Framingham Omni 1 participants (*n* = 285) who attended their third examination (2007–2008). Participants were excluded in a hierarchical order if they were missing RBC fatty acid measurements (*n* = 143), biomarker measurements (*n* = 342), or clinical covariates (*n* = 12). The study protocol was approved by the Institutional Review Board of the Boston University Medical Center. Informed consent was provided by all participants.

### 2.2. Red Blood Cell Linoleic Acid and Arachidonic Acid

Blood was drawn after a 10–12 h fast into an EDTA tube, and RBCs were separated from plasma by centrifugation. The RBC fraction was frozen at −80 °C immediately after collection. RBC fatty acid composition was determined as described previously (OmegaQuant Analytics, Sioux Falls, SD, USA) [29]; it is known to be more stable compared to plasma FAs [30], representing the last ~4 months of nutritional, metabolic environments and tissue membrane FA profiles [31]. Briefly, RBCs were incubated at 100 °C with boron trifluoride-methanol and hexane to generate fatty acid methyl esters that were then analyzed by gas chromatography with flame ionization detection. The coefficients of variation were 5.3% for LA and 2.3% for AA.

### 2.3. Inflammatory Biomarkers

We selected one urinary and nine serum biomarkers representing multiple inflammatory pathways: urinary 8-epi-PGF2α isoprostanes (normalized to creatinine), *C*-reactive protein (CRP), interleukin-6, intercellular adhesion molecule-1 (ICAM-1), lipoprotein-associated phospholipase-A2 (Lp-PLA2) activity and mass, monocyte chemoattractant protein-1 (MCP-1), osteoprotegerin, P-selectin, and tumor necrosis factor receptor-2 (TNFR2) (Table 1). The details of the rational for selection of these biomarkers, assays, and measurements have been described previously [32]. Briefly, test kits used for quantification were ACE Competitive EIA (Cayman Chemical, Ann Arbor, MI, USA) for 8-epi-PGF2α isoprostanes; BN100 nephelometer for CRP (Dade Behring, Deerfield, IL, USA); Quantitative ELISA (R&D systems, Minneapolis, MN, USA) for interleukin-6, ICAM-1, MCP-1, P-selectin, and TNFR2; Quantitative ELISA (diaDexus, South San Francisco, CA, USA) for LpPLA2 activity and mass; and Quantitative ELISA (Biomedica Gesellschaft mbH, Vienna, Austria) for osteoprotegerin [32]. The inter-assay coefficients of variation were less than 10% for all measurements [18,32].

### 2.4. Statistical Analysis

Descriptive statistics are presented as percentage for categorical variables and as mean ± standard deviation for continuous variables. To normalize skewed distributions, the analyses of inflammatory biomarkers were natural logarithmically transformed. The relationships of RBC LA and AA and the logarithmic values of inflammatory biomarkers were evaluated using partial correlation coefficients. In addition, the associations between these biomarkers and RBC *n*-3 PUFA EPA + DHA, i.e., the omega-3 index (O3I), originally reported in Fontes et al. [25], were re-evaluated controlling for AA and LA levels, to confirm that the relationships originally observed remained after adjustment for these additional PUFAs. We conducted a series of multivariable models adjusting for different subsets of subject characteristic variables and FA exposures: Model 1 adjusted for age and sex, Model 2 adjusted for all Table 2 demographic covariates, and Model 3 included Model 2 covariates + LA or AA (mutual adjustment) + O3I. Statistical significance was defined by two-tailed *p* < 0.05 for each FA vs. biomarker comparison. We also looked for the evidence of non-linearity or statistical interaction (age, sex, race/ethnicity) in Model 3. Given the exploratory nature of the analyses, we used a Bonferroni corrected statistical significance threshold of 0.005 (0.05/10 inflammatory markers) for tests of non-linearity and statistical interaction. Tests of non-linearity were conducted by fitting a cubic spline to the FA exposure and comparing that model using a nested F-test to the same model with a linear term for the FA exposure. Tests of interaction were conducted by evaluating the statistical significance of an interaction term between the FA exposure and (separately) age, sex, or race/ethnicity (non-Hispanic White vs. other). All statistical analyses were performed using the R Statistical Software (v4.2.0; R Core Team 2022).

## 3. Results

### 3.1. Descriptive Statistics

We evaluated 2777 eligible participants from the Framingham Offspring and Omni cohorts. Their mean age was 66 years and 54% were female (Table 2). The intercorrelations between RBC LA, AA, and the O3I are in the footnote of Table 1, while additional descriptive statistics for RBC LA, AA, and the O3I are reported in Appendix A.

### 3.2. The Correlations Between Linoleic Acid, Arachidonic Acid, and Inflammatory Biomarkers

The relationships between the RBC LA levels (after accounting for variation in inflammatory markers accounted for by demographic and medical history variables, as well as AA and O3I; Model 3) were statistically significant and inverse for five markers: CRP, IL-6, ICAM-1, MCP-1, and P-selectin (Table 3). There were no inflammatory biomarkers that were positively and significantly associated with RBC LA after removing the association of AA and the O3I with these biomarkers.

Similarly, the relationships of the RBC AA levels (after accounting for variation in inflammatory markers for all other variables, including LA and O3I; Model 3) were statistically significant and inverse for four markers: IL-6, ICAM-1, MCP-1, and osteoprotegerin (Table 4). There were no inflammatory biomarkers that were positively and significantly associated with RBC AA after removing the influence of LA and the O3I on these biomarkers.

Similar analyses as the above one were conducted using the O3I as the exposure in order to confirm that the previously published results by Fontes et al. [25] were not altered by additional adjustment for LA and AA (which was not reported in the original paper). The original findings were confirmed with all marker associations being significant (*p* < 0.01), except the inverse relationship with MCP-1; although still inverse, it was no longer statistically significant (*r* = −0.038, *p* = 0.06; Appendix A).

### 3.3. Tests of Non-Linearity and Interaction

One FA-inflammatory marker relationship showed some evidence of non-linearity, i.e., AA and osteoprotegerin (*p* = 0.0005), with Q4/Q5 of AA levels showing much stronger inverse associations with osteoprotegerin than Q1-Q3 (see Appendix A). There was no evidence of statistically significant interaction with age (continuous) or sex (male vs. female), though for one FA, there was a statistically significant interaction with race (*p* = 0.001). ICAM-1 showed stronger inverse association with AA levels in non-Whites and Hispanics [95% CI (−0.85, −0.32); *n* = 266], as compared to non-Hispanic Whites [95% CI, (−0.16, −0.05); *n* = 2490]; see Appendix A).

## 4. Discussion

In our large community-based study, RBC LA was inversely associated with CRP, IL-6, ICAM-1, MCP-1, and P-selectin, while RBC AA was inversely associated with IL-6, ICAM-1, MCP-1, and osteoprotegerin, all after adjusting for multiple confounders including mutual *n*-6 PUFA biomarker adjustments and the O3I. Three of these in common are IL-6, ICAM-1, and MCP-1. Importantly, neither of these *n*-6 PUFAs was significantly and positively associated with any of the ten inflammatory biomarkers.

Several previous studies have examined the relation between the dietary intake of *n*-6 PUFA and blood levels of *n*-6 PUFA to inflammatory biomarkers. Among the dietary studies [62,63,64,65], three of four found a significant inverse association between *n*-6 PUFA intake and CRP [63,64,65], while all ten biomarker-based studies reported significant inverse relations for at least one inflammatory marker with total *n*-6, LA, or AA levels [66,67,68,69,70,71,72,73,74,75,76]. Higher LA is most commonly reported to be associated with lower levels of CRP [63,64,65,67,68,71,72,73,74], though two studies reported links with higher Lp-PLA2 [69,75]. Findings for AA tend to be mixed, with mostly no significant associations with inflammatory biomarkers [63,64,71,73,74], but inverse associations with IL-6 [66] and Lp-PLA2 mass and activity [69,75]. Discrepancies between studies could often be attributed to differences in populations, lipid compartment analyzed (e.g., plasma vs. RBC), assay methods and the time of blood drawn for both FAs and the inflammatory markers. Our findings extend the available evidence by additionally exploring a larger number of inflammatory biomarkers representing major inflammatory mechanisms. However, it is important to note that correlations were mostly weak, even when statistically significant.

*n*-6 PUFA metabolism is closely linked to *n*-3 PUFAs. In conjunction, their metabolites affect multiple pathways pleiotropically and may result in the production of both anti-inflammatory and pro-inflammatory mediators [77]. In the context of *n*-6 PUFAs, LA and AA metabolites, i.e., oxylipins and eicosanoids produced via cyclooxygenase (COX), 5-lipoxygenase (5-LOX), 12-LOX, and 15-LOX pathways [77], exhibit pro-aggregatory effects, e.g., thromboxanes, as well as anti-aggregatory, anti-inflammatory, and vasodilatory effects, e.g., epoxy-eicosatrienoic acids, prostaglandin E2, which induces IL-6 production, lipoxin A4, and prostacyclin. An 8-week supplementation of soy oil (~50% LA) elevated the levels of Lp-PLA2 activity in healthy Korean adults [78]. However, the meta-analyses of randomized controlled trials found no significant effect of higher LA intake on a wide range of inflammatory biomarkers, including CRP, IL-6, ICAM-1, P-selectin, and MCP-1 [18,79]. Similarly, the levels of pro-inflammatory metabolites did not increase with AA supplementation in human trials [23,24,80,81], even though experimental studies suggest that AA elevates the expression of ICAM-1 [82] and increases osteoclastogenesis [83]. In observational studies, LA is inversely associated with mortality, cancer, T2DM, and other types of death [19,20,21,84,85], while associations between AA and CVD [86], cancer [85], diabetes [20], atrial fibrillation [87], and death [88] remain generally neutral. Collectively, based on the above evidence, our findings do not support the hypothesis that *n*-6 PUFAs are pro-inflammatory.

Our findings, although observational, do not establish causality. Instead, they suggest that RBC *n*-6 PUFAs are not pro-inflammatory (refuting at least one of the “concerns” of seed oils). Based on several biomarkers, they actually have an anti-inflammatory signature. Even though correlations are not as strong as established risk factors, e.g., BMI (*r =* 0.32 with CRP) [89], they are comparable to strong predictors of obesity, such as pericardial fat and intrathoracic fat [90]. LA is an essential fatty acid. Recommendations to reduce it in the diet could lead to an inadequate intake resulting, in the extreme, in an essential fatty acid deficiency manifested by dermatitis, alopecia, poor wound healing [91], and growth stunting in children [92]. Modeling exercises studying the potential effects of LA intake from “high mono” seed oils (e.g., safflower and sunflower hybrids) on a population level have raised concerns that the health benefits of LA (beyond preventing classic deficiency symptoms) may be threatened by the increased use of these low LA oils [93]. Thus, our findings support the current dietary guidelines [94] to prioritize LA-rich oils, including soybean, corn, and sunflower oils, and to decrease saturated fat intake (i.e., palmitic acid, stearic acid, and predominant FAs in beef tallow or butter).

The independent effects of LA and AA and their metabolites on the less-studied Lp-PLA2, urinary isoprostanes, osteoprotegerin, and TNFR2 leave much room for further investigation. Such studies could explore the role of LA or AA in the management of chronic inflammatory conditions in patients versus healthy subjects, using higher doses and/or longer treatment periods, and track the effects of change in RBC *n*-6 PUFAs on inflammatory biomarkers over time.

The strengths of this study include the use of a well-characterized and large community-based cohort with the rigorous ascertainment of clinical risk factors, inflammatory biomarkers, and individual RBC LA and AA levels. The inclusion of the minority Framingham Omni cohort in addition to the Offspring cohort improves the generalizability of the findings. To our knowledge, our study is one of the largest to examine the relationships between circulating individual RBC *n*-6 PUFA levels and inflammatory markers. The present study is also unique in including up to ten different inflammatory biomarkers associated with widely varying pathways and phases of inflammation.

This study also has limitations. Observational study, especially cross-sectional ones, cannot ascribe causal connections between RBC *n*-6 PUFAs and inflammatory markers, nor can they exclude the possibility of residual confounding. Inferences on if RBC *n*-6 PUFAs determinately contribute to a meaningfully lower inflammatory state would require further study. As noted, the correlation coefficients, although statistically significant and inverse, were nevertheless low. Hence, the clinical relevance of these findings is unclear. However, the polyunsaturated fatty acid levels tend to correlate well over significant periods of repeated measurements [95], lending some support to theorized longitudinal effects, worthy of future investigations. Participants were mainly middle-aged to older adults from Framingham, Massachusetts. Thus, our findings may not necessarily be representative of individuals that are younger or from other geographic areas.

## 5. Conclusions

Our community-based study identified small, significant, inverse associations between the RBC LA and AA levels and six major biomarkers of inflammation (three in common: IL-6, ICAM-1, and MCP-1), representing a wide variety of inflammation pathways. Our results suggest that LA is more likely to be anti- than pro-inflammatory, and the present efforts to reduce its intake are ill advised.

## Figures and Tables

**Table 1 nutrients-17-02076-t001:** The summary of the examined inflammatory markers (all are serum or plasma markers unless otherwise specified).

Inflammatory Marker	Units	Role in Inflammation
Urinary 8-EPI-Isoprostanes/Creatinine	0.18–0.40 μg/g creatinine [33]	A stable biomarker of oxidative stress and inflammation formed from non-enzymatic free radical-catalyzed peroxidation of AA [34]
*C*-Reactive Protein	≤3 mg/L [35]	An acute-phase protein produced in the liver; levels rise rapidly and acutely during inflammation [36]
Interleukin-6	1–5 pg/mL [37]	A pleotropic cytokine that exhibits both anti-inflammatory and pro-inflammatory effects [38,39]; an essential component for CRP production via hepatocytes [40].
Intercellular Adhesion Molecule 1	100–300 ng/mL [41,42]	A cell surface glycoprotein that mediates and facilitates leukocytes to sites of inflammation [43]; levels are upregulated during inflammation [44]
Lp-PLA2 Activity	225 nmol/min/mL [45,46]	A pro-inflammatory enzyme from the phospholipase A2 family; a potential marker of vascular inflammation [47,48]
Lp-PLA2 Concentration	≤200 ng/mL [49]	
Monocyte Chemoattractant Protein-1	127–274 pg/mL [50] ^1^	A protein that is upregulated by pro-inflammatory stimuli; attracts monocytes, neutrophils, and lymphocytes to sites of inflammation [51]
Osteoprotegerin	13–84 pg/mL [52] ^1^	A pro-inflammatory soluble decoy receptor and a member of the TNF receptor that potentially acts through NF-κB activation [53]; inhibits bone resorption by preventing RANKL from engaging RANK receptors [54]
P-Selectin	19–521 ng/mL [55]	A member of the selectin adhesion molecule expressed on activated endothelial cells and platelets; facilitates and mediates leukocytes at inflammation sites [56]
Tumor Necrosis Factor Receptor 2	1951–3430 pg/mL [57,58,59] ^1^	A cytokine that exhibits both anti-inflammatory (e.g., neuroprotection) [60] and pro-inflammatory effects (e.g., promotes tumor cell proliferation) [61]

Lp-PLA2 = lipoprotein-associated phospholipase-A2. ^1^ Not absolute, levels may vary due to age, genetics, and overall health.

**Table 2 nutrients-17-02076-t002:** The participant characteristics from the Framingham Study.

Characteristics	% or Mean ± SD
Sex (% female)	54.1
Age (years)	65.9 ± 9.0
Race/ethnicity	
Non-Hispanic White (%)	89.7
NH Black (%)	3.5
NH Asian (%)	2.3
NH Other (%)	0.7
Hispanic (%)	3.1
Current Smoker (%)	7.3
Systolic Blood Pressure (mmHg)	129 ± 17
Body Mass Index (kg/m^2^)	28.4 ± 5.5
Total Cholesterol (mg/dL)	186 ± 37
HDL Cholesterol (mg/dL)	57 ± 18
Triglycerides (mg/dL)	117 ± 68
Glucose (mg/dL)	107 ± 24
Aspirin Usage (% reporting ≥3 times a week)	43.6
Prevalent Dyslipidemia Medication (%)	45.1
Prevalent Hypertension Medication (%)	49.7
Prevalent Diabetes (%)	13.6
Prevalent Cardiovascular Disease (CVD) (%)	16.0
Hormone Replacement Therapy (%)	13.4
Exposures ^1^	
RBC Linoleic Acid (LA, %)	11.04 ± 1.71
RBC Arachidonic Acid (AA, %)	16.57 ± 1.60
Omega-3 Index (O3I, RBC EPA + DHA, %)	5.57 ± 1.71
Outcomes	
Isoprostanes/Creatinine (mg/mg)	11.2 ± 6.2
*C*-Reactive Protein (mg/L)	3.2 ± 7.3
Interleukin-6 (pg/mL)	2.6 ± 3.0
Intercellular Adhesion Molecule 1 (ng/mL)	294.4 ± 104.4
Lp-PLA2 Activity (nmol/min/mL)	137.4 ± 34.9
Lp-PLA2 Mass (ng/mL)	199.5 ± 49.7
Monocyte Chemoattractant Protein-1 (pg/mL)	381.5 ± 131.3
Osteoprotegerin (pmol/L)	5.0 ± 1.6
P-Selectin (ng/mL)	41.2 ± 13.2
Tumor Necrosis Factor Receptor-2 (pg/mL)	2591.1 ± 1055.1

Lp-PLA2 = lipoprotein-associated phospholipase-A2. ^1^ The correlation between RBC LA% and AA% was −0.47; between LA% and the O3I was −0.23, and between AA% and the O3I was −0.40.

**Table 3 nutrients-17-02076-t003:** The partial correlations (95% CI) between RBC LA and inflammatory markers in the Framingham Study with differing levels of covariate adjustment.

	Model 1	Model 2	Model 3
Isoprostanes/Creatinine	−0.025 (−0.062, 0.013)	0.004 (−0.036, 0.044)	−0.048 (−0.102, 0.005)
CRP	**−0.063 (−0.101, −0.026) ****	**−0.042 (−0.079, −0.005) ***	**−0.061 (−0.111, −0.011) ***
Interleukin-6	**−0.105 (−0.141, −0.069) ****	**−0.056 (−0.093, −0.019) ****	**−0.146 (−0.195, −0.096) ****
ICAM-1	−0.035 (−0.073, 0.003)	0.014 (−0.024, 0.052)	**−0.088 (−0.139, −0.037) ****
LpPLA2 Activity	**0.099 (0.063, 0.135) ****	**0.046 (0.014, 0.078) ****	0.034 (−0.010, 0.077)
LpPLA2 Mass	**0.117 (0.080, 0.155) ****	**0.042 (0.004, 0.079) ***	0.027 (−0.025, 0.078)
MCP-1	−0.030 (−0.068, 0.007)	−0.020 (−0.060, 0.020)	**−0.068 (−0.122, −0.015) ***
Osteoprotegerin	**0.060 (0.026, 0.093) ****	**0.071 (0.035, 0.106) ****	0.015 (−0.033, 0.063)
P-selectin	**−0.038 (−0.075, −0.000) ***	−0.037 (−0.077, 0.003)	**−0.067 (−0.121, −0.013) ***
TNFR2	−0.017 (−0.053, 0.019)	0.018 (−0.020, 0.056)	−0.028 (−0.080, 0.024)

*** *p* < 0.05; ** *p* < 0.01**; Model 1 = age and sex only; Model 2 = all Table 2 variables (except fatty acids); Model 3 = all Table 2 variables + AA + O3I.

**Table 4 nutrients-17-02076-t004:** The partial correlations (95% CI) between RBC AA and inflammatory markers in the Framingham Study with differing levels of covariate adjustment.

	Model 1	Model 2	Model 3
Isoprostanes/Creatinine	**0.069 (0.032, 0.106) ****	**0.048 (0.010, 0.086) ***	−0.037 (−0.093, 0.019)
CRP	**0.093 (0.056, 0.130) ****	**0.066 (0.031, 0.101) ****	0.003 (−0.049, 0.055)
Interleukin-6	**0.094 (0.058, 0.129) ****	**0.047 (0.012, 0.083) ****	**−0.096 (−0.147, −0.044) ****
ICAM-1	−0.010 (−0.047, 0.027)	−0.034 (−0.071, 0.002)	**−0.137 (−0.190, −0.084) ****
LpPLA2 Activity	**−0.059 (−0.094, −0.023) ****	−0.000 (−0.031, 0.031)	−0.004 (−0.049, 0.041)
LpPLA2 Mass	**−0.044 (−0.081, −0.006) ***	0.023 (−0.013, 0.060)	0.004 (−0.049, 0.058)
MCP-1	0.002 (−0.035, 0.038)	0.006 (−0.032, 0.044)	**−0.060 (−0.116, −0.004) ***
Osteoprotegerin	−0.029 (−0.062, 0.004)	**−0.044 (−0.078, −0.010) ***	**−0.074 (−0.124, −0.024) ****
P-selectin	0.023 (−0.014, 0.060)	**0.053 (0.015, 0.091) ****	−0.017 (−0.073, 0.039)
TNFR2	−0.017 (−0.053, 0.019)	0.018 (−0.020, 0.056)	−0.028 (−0.080, 0.024)

*** *p* < 0.05; ** *p* < 0.01**; Model 1 = age and sex only; Model 2 = all Table 1 variables (except fatty acids); Model 3 = all Table 2 variables + LA + O3I.

## Data Availability

This manuscript was prepared using FRAMOFFSPRING Research Materials obtained from the NHLBI Biologic Specimen and Data Repository Information Coordinating Center and does not necessarily reflect the opinions or views of the FRAMOFFSPRING or the NHLBI. The data presented in this study are not available on request from the corresponding author due to privacy and data agreements, but are available via the original study investigators by application through BIOLINCC or other publicly available repositories.

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
