# Peer review of "Red Blood Cell Omega-6 Fatty Acids and Biomarkers of Inflammation in the Framingham Offspring Study"

_nutrients, 2025, doi:10.3390/nu17132076_

Round 1
Reviewer 1 Report
Comments and Suggestions for Authors
The paper "Red Blood Cell Omega-6 Fatty Acids and Biomarkers of Inflammation in the Framingham Offspring Study" adds a small piece of knowledge regarding the activity of some lipids at the cardiovascular level. It is a well-written work, not very innovative and elaborate but still well structured to present the data that emerged from the experimental sessions. In my opinion, it can be published in this version, but I suggest just one small change:
Line 250 "they suggest that n-6 PUFAs are not pro-inflammatory": I would specify that they showed this behaviour in the context of RBC membrane lipids, because it is not certain that in another experimental system the same thing would happen.
Author Response
Reviewer 1:
The paper "Red Blood Cell Omega-6 Fatty Acids and Biomarkers of Inflammation in the Framingham Offspring Study" adds a small piece of knowledge regarding the activity of some lipids at the cardiovascular level. It is a well-written work, not very innovative and elaborate but still well structured to present the data that emerged from the experimental sessions. In my opinion, it can be published in this version, but I suggest just one small change:
Line 250 "they suggest that n-6 PUFAs are not pro-inflammatory": I would specify that they showed this behaviour in the context of RBC membrane lipids, because it is not certain that in another experimental system the same thing would happen.
Thank you for your comment. To reflect your inputs, we have amended this in the following sections:
Methods
“RBC fatty acid composition was determined as described previously (OmegaQuant Analytics, Sioux Falls, SD, USA) [1], and are known to be more stable vs. plasma FAs [2], representing the last ~4 months of nutritional, metabolic environments, and tissue membrane FA profiles [3].” (Line 112-115)
Discussion
“Our findings, although observational, do not establish causality. Instead, they suggest that RBC n-6 PUFAs…” (Line 255-256)
References
- Harris, W.S.; Pottala, J.V.; Vasan, R.S.; Larson, M.G.; Robins, S.J. Changes in erythrocyte membrane trans and marine fatty acids between 1999 and 2006 in older Americans. J Nutr 2012, 142, 1297-1303, doi:10.3945/jn.112.158295.
- Harris, W.S.; Pottala, J.V.; Sands, S.A.; Jones, P.G. Comparison of the effects of fish and fish-oil capsules on the n 3 fatty acid content of blood cells and plasma phospholipids. Am J Clin Nutr 2007, 86, 1621-1625, doi:10.1093/ajcn/86.5.1621.
- Fenton, J.I.; Gurzell, E.A.; Davidson, E.A.; Harris, W.S. Red blood cell PUFAs reflect the phospholipid PUFA composition of major organs. Prostaglandins Leukot Essent Fatty Acids 2016, 112, 12-23, doi:10.1016/j.plefa.2016.06.004.

Reviewer 2 Report
Comments and Suggestions for Authors
This study is a well-conceived and meticulously articulated investigation that explores a pertinent and contentious issue within the field of nutrition science, the influence of omega-6 polyunsaturated fatty acids (PUFAs) on inflammation. The researchers utilized a large, well-characterized community-based cohort, specifically the Framingham Offspring and Omni cohorts, to examine the cross-sectional relationships between red blood cell (RBC) concentrations of linoleic acid (LA) and arachidonic acid (AA) and a comprehensive array of ten inflammatory biomarkers. The study's principal strengths include its substantial sample size, the employment of objectively measured RBC fatty acid levels—which serve as indicators of long-term intake and metabolism—the rigorous and transparent statistical analysis incorporating progressive multivariable adjustments, and a nuanced discussion that thoughtfully engages with existing literature. The discovery that both LA and AA exhibit weak yet statistically significant associations with lower, rather than higher, levels of several inflammatory markers represents a significant contribution to the field, directly challenging the prevailing hypothesis that omega-6 PUFAs are pro-inflammatory. The manuscript is well-structured, and the conclusions are generally substantiated by the data presented. It is potentially suitable for publication in Nutrients following attention to the major and minor points detailed below.
Major points
- The primary concern is the clarity of the results reporting in Section 3.2 and its interpretation. The authors present four statistical models with increasing levels of covariate adjustment, which is a major strength. However, the narrative description of the results does not always clearly specify which model is being referred to, and it fails to discuss the implications of the changes in associations across the models. For instance, in Section 3.2, the text states that RBC LA was inversely correlated with five markers, but a look at Table 3 shows a complex pattern. The association between LA and C-Reactive Protein (CRP) is significant in Model 2 but becomes non-significant in Model 3 (after adjusting for AA) and then significant again in Model 4 (after further adjusting for the Omega-3 Index). This suggests a complex interplay or statistical suppression effect between LA, AA, and omega-3 PUFAs. This is a critical finding that is currently overlooked in the discussion. The authors should explicitly address these dynamics. Why does adjusting for AA nullify the LA-CRP association, and why does subsequent adjustment for the O3I restore it? A thorough discussion of the intercorrelations between these fatty acids and how they influence the regression models is necessary to properly interpret the "independent" associations and would significantly strengthen the manuscript.
- The authors correctly acknowledge in the limitations that the observed correlations, while statistically significant, are weak (e.g., partial correlation coefficients range from approximately -0.06 to -0.15). This raises questions about the clinical or biological relevance of the findings. While the large sample size provides the statistical power to detect such small effects, the manuscript would be improved by a more in-depth discussion of this point. The authors could attempt to contextualize the magnitude of these associations by comparing them to the effects of other known determinants of inflammation (e.g., BMI, smoking) within their cohort or from the literature. While the study's primary aim is to test the pro-inflammatory hypothesis rather than to establish a clinical intervention, a brief discussion on whether these small effects could, in aggregate or over a lifetime, contribute to a meaningfully lower inflammatory state would add valuable perspective for the reader.
Minor Points
- Table 1 (Inflammatory Markers). The units for "Urinary 8-EPI-Isoprostanes/Creatinine" are listed as "μg/g creatinine" in the main table, but footnote 1 states it was "Reported as pg/mmol in the Framingham Heart Study". For consistency and clarity, the authors should use consistent units or clarify the conversion.
- Clarity in Abstract: In the abstract, the sentence listing the findings for LA reads: "...monocyte chemoattractant protein-1, and (r=-0.07); P-selectin (r=-0.07)." The comma and "and" after MCP-1 are grammatically awkward and should be revised for better flow.
- The first sentence of the Conclusion (Section 5) states that the study identified inverse associations between RBC LA and AA levels with "six major biomarkers of inflammation". The abstract and discussion correctly detail that LA was associated with five unique markers and AA with four, with three of these being common to both. This does indeed total six unique biomarkers (CRP, P-selectin, IL-6, ICAM-1, MCP-1, Osteoprotegerin). However, the way it is stated could be made clearer earlier in the discussion to avoid any potential reader confusion.
- Duplicate References. For instance, refs 7 and 8.
Author Response
Reviewer 2:
Major points
The primary concern is the clarity of the results reporting in Section 3.2 and its interpretation. The authors present four statistical models with increasing levels of covariate adjustment, which is a major strength. However, the narrative description of the results does not always clearly specify which model is being referred to, and it fails to discuss the implications of the changes in associations across the models. For instance, in Section 3.2, the text states that RBC LA was inversely correlated with five markers, but a look at Table 3 shows a complex pattern. The association between LA and C-Reactive Protein (CRP) is significant in Model 2 but becomes non-significant in Model 3 (after adjusting for AA) and then significant again in Model 4 (after further adjusting for the Omega-3 Index). This suggests a complex interplay or statistical suppression effect between LA, AA, and omega-3 PUFAs. This is a critical finding that is currently overlooked in the discussion. The authors should explicitly address these dynamics. Why does adjusting for AA nullify the LA-CRP association, and why does subsequent adjustment for the O3I restore it? A thorough discussion of the intercorrelations between these fatty acids and how they influence the regression models is necessary to properly interpret the "independent" associations and would significantly strengthen the manuscript.
Thank you for your critical analysis. We acknowledge that the correlation differences between Models 2 and 3 are interesting, and there might be unknown mechanisms or relationships at play between LA, AA, O3I and the inflammatory markers. The intercorrelations between these FAs are reported in the Table 2 footnote, where r = -0.47 for LA% and AA%; r = -0.23 for LA% and the Omega-3 index, (O3I), and r = -0.40 for AA% and the O3I – neither suggest any potential opposite directions that may lead to the differences in the model. This makes the phenomenon difficult to explain as it requires further investigation and may unintendedly distract the reader from our main message. As our main findings are based on Model 4, we have simplified our findings to re-focus the discussion to the primary model by removing Model 3 (which is an intermediate model and only adjusts for AA). The following adjustments to the text are made to accommodate this change:
Methods
“…Model 3 included Model 2 covariates + LA or AA (mutual adjustment) + O3I.” (Line 148-150)
Results
Table(s) 3 & 4: Model 3 findings are removed, and Model 4 findings are renamed as Model 3, with appropriate changes to the footnotes.
“Similarly, the relationships of RBC AA levels (after accounting for variation in inflammatory markers for all other variables, including LA and O3I; Model 3) were statistically significant…” (Line 186-188)
The authors correctly acknowledge in the limitations that the observed correlations, while statistically significant, are weak (e.g., partial correlation coefficients range from approximately -0.06 to -0.15). This raises questions about the clinical or biological relevance of the findings. While the large sample size provides the statistical power to detect such small effects, the manuscript would be improved by a more in-depth discussion of this point. The authors could attempt to contextualize the magnitude of these associations by comparing them to the effects of other known determinants of inflammation (e.g., BMI, smoking) within their cohort or from the literature. While the study's primary aim is to test the pro-inflammatory hypothesis rather than to establish a clinical intervention, a brief discussion on whether these small effects could, in aggregate or over a lifetime, contribute to a meaningfully lower inflammatory state would add valuable perspective for the reader.
Thank you for your insights. We believe the main focus of our findings are that LA and AA is not pro-inflammatory. The correlation coefficients from our study are generally comparable to those between pericardial fat, intrathoracic fat (good predictors of obesity) and inflammatory markers in the same study [1]. Alongside comparisons with determinants that are more established, this is relatively weaker. In an older Chinese cohort, BMI is weakly correlated with CRP (r = 0.32) [2]. Though, it is important to point out that neither of the studies adjust for covariates extensively as we did.
Polyunsaturated fatty acid levels tend to correlate well over time. E.g., pairwise correlations for plasma FAs in the Cardiovascular Health Study measured six and thirteen years apart for LA range between r = 0.50-0.65, and for AA r = 0.58-0.76, similar to published correlations for omega-3 FAs [3]. As RBC FAs are more stable, we would expect correlations to be stronger. This lends some support to suggest that any theorised effects may be longitudinal. However, the question if it may determinately contribute to a meaningfully lower inflammatory state would require further investigation using time-to-event survival analyses, which is beyond the scope of the current study, but valuable to follow through in the future.
We have added the relevant points of our response to the discussion:
“Even though correlations are not as strong as established risk factors, e.g., BMI (r = 0.32 with CRP) [2], they are comparable to strong predictors of obesity, such as pericardial fat and intrathoracic fat [1].” (Line 258-260)
“Inferences on if RBC n-6 PUFAs determinately contribute to a meaningfully lower inflammatory state would require further study. As noted, the correlations coefficients, although statistically significant and inverse, were nevertheless low. Hence, the clinical relevance of these findings is unclear. However, polyunsaturated fatty acid levels tend to correlate well over significant periods of repeated measurements [3], lending some support to theorized longitudinal effects, worthy of future investigations.” (Line 287-293)
Minor Points
Table 1 (Inflammatory Markers). The units for "Urinary 8-EPI-Isoprostanes/Creatinine" are listed as "μg/g creatinine" in the main table, but footnote 1 states it was "Reported as pg/mmol in the Framingham Heart Study". For consistency and clarity, the authors should use consistent units or clarify the conversion.
We recognize that it is confusing to draw attention to different units and have now removed the reference to maintain consistency.
Clarity in Abstract: In the abstract, the sentence listing the findings for LA reads: "...monocyte chemoattractant protein-1, and (r=-0.07); P-selectin (r=-0.07)." The comma and "and" after MCP-1 are grammatically awkward and should be revised for better flow.
We have altered this to: “…monocyte chemoattractant protein-1 (r=-0.07); and P-selectin (r=-0.07). (Line 23)
The first sentence of the Conclusion (Section 5) states that the study identified inverse associations between RBC LA and AA levels with "six major biomarkers of inflammation". The abstract and discussion correctly detail that LA was associated with five unique markers and AA with four, with three of these being common to both. This does indeed total six unique biomarkers (CRP, P-selectin, IL-6, ICAM-1, MCP-1, Osteoprotegerin). However, the way it is stated could be made clearer earlier in the discussion to avoid any potential reader confusion.
To clarify this, we have added the following to Section 4 and 5:”
“Three of these in common are IL-6, ICAM-1, and MCP-1.” (Line 217-218)
“Our community-based study identified small, significant, inverse associations between RBC LA and AA levels with six major biomarkers of inflammation (three in common: IL-6, ICAM-1, MCP-1)…” (Line 299-300)
Duplicate References. For instance, refs 7 and 8.
Thank you for spotting this. We have now corrected this without the duplicate reference: “…risk for depression [4], CVD [5,6], and total mortality [5,6] bringing into question their cardioprotective reputation [6].” (Line 45-46)
References
- Tadros, T.M.; Massaro, J.M.; Rosito, G.A.; Hoffmann, U.; Vasan, R.S.; Larson, M.G.; Keaney, J.F., Jr.; Lipinska, I.; Meigs, J.B.; Kathiresan, S., et al. Pericardial fat volume correlates with inflammatory markers: the Framingham Heart Study. Obesity (Silver Spring, Md.) 2010, 18, 1039-1045, doi:10.1038/oby.2009.343.
- Ye, X.; Yu, Z.; Li, H.; Franco, O.H.; Liu, Y.; Lin, X. Distributions of C-Reactive Protein and its Association With Metabolic Syndrome in Middle-Aged and Older Chinese People. J Am Coll Cardiol 2007, 49, 1798-1805, doi:https://doi.org/10.1016/j.jacc.2007.01.065.
- Lai, H.T.; de Oliveira Otto, M.C.; Lemaitre, R.N.; McKnight, B.; Song, X.; King, I.B.; Chaves, P.H.; Odden, M.C.; Newman, A.B.; Siscovick, D.S., et al. Serial circulating omega 3 polyunsaturated fatty acids and healthy ageing among older adults in the Cardiovascular Health Study: prospective cohort study. BMJ 2018, 363, k4067, doi:10.1136/bmj.k4067.
- Yin, J.; Li, S.; Li, J.; Gong, R.; Jia, Z.; Liu, J.; Jin, Z.; Yang, J.; Liu, Y. Association of serum oleic acid level with depression in American adults: a cross-sectional study. BMC Psychiatry 2023, 23, 845, doi:10.1186/s12888-023-05271-0.
- Steffen, B.T.; Duprez, D.; Szklo, M.; Guan, W.; Tsai, M.Y. Circulating oleic acid levels are related to greater risks of cardiovascular events and all-cause mortality: The Multi-Ethnic Study of Atherosclerosis. Journal of Clinical Lipidology 2018, 12, 1404-1412, doi:https://doi.org/10.1016/j.jacl.2018.08.004.
- Lai, H.T.M.; de Oliveira Otto, M.C.; Lee, Y.; Wu, J.H.Y.; Song, X.; King, I.B.; Psaty, B.M.; Lemaitre, R.N.; McKnight, B.; Siscovick, D.S., et al. Serial Plasma Phospholipid Fatty Acids in the De Novo Lipogenesis Pathway and Total Mortality, Cause‐Specific Mortality, and Cardiovascular Diseases in the Cardiovascular Health Study. J Am Heart Assoc 2019, 8, e012881, doi:10.1161/JAHA.119.012881.
